# Research on control strategy of pneumatic soft bionic robot based on improved CPG

**Wenchuan Zhao**[1]*, **Yu Zhang**[2], **Kian Meng Lim**[3], **Lijian Yang**[1], **Ning Wang**[2], **Linghui Peng**[2]

1 School of Information Science and Engineering, Shenyang University of Technology, Shenyang, China, 2 School of Mechanical Engineering, Shenyang University of Technology, Shenyang, China, 3 Department of Mechanical Engineering, National University of Singapore, Singapore, Singapore

* zhao_wenchuan@126.com

**Data Availability Statement:** All relevant data are within the manuscript and its Supporting information files.

## Abstract

To achieve the accuracy and anti-interference of the motion control of the soft robot more effectively, the motion control strategy of the pneumatic soft bionic robot based on the improved Central Pattern Generator (CPG) is proposed. According to the structure and motion characteristics of the robot, a two-layer neural network topology model for the robot is constructed by coupling 22 Hopfield neuron nonlinear oscillators. Then, based on the Adaptive Neuro-Fuzzy Inference System (ANFIS), the membership functions are offline learned and trained to construct the CPG-ANFIS-PID motion control strategy for the robot. Through simulation research on the impact of CPG-ANFIS-PID input parameters on the swimming performance of the robot, it is verified that the control strategy can quickly respond to input parameter changes between different swimming modes, and stably output smooth and continuous dynamic position signals, which has certain advantages. Then, the motion performance of the robot prototype is analyzed experimentally and compared with the simulation results. The results show that the CPG-ANFIS-PID motion control strategy can output coupled waveform signals stably, and control the executing mechanisms of the pneumatic soft bionic robot to achieve biological rhythms motion propulsion waveforms, confirming that the control strategy has accuracy and anti-interference characteristics, and enable the robot have certain maneuverability, flexibility, and environmental adaptability. The significance of this work lies in establishing a CPG-ANFIS-PID control strategy applicable to pneumatic soft bionic robot and proposing a rhythmic motion control method applicable to pneumatic soft bionic robot.

## 1 Introduction

Soft robots have infinite degrees of freedom, while the number of actuators is limited, making it difficult to control the shape and end position of the robot in real-time with precision [1, 2]. Typically, a finite-dimensional model is used to describe the distribution parameter model with infinite dimensions, and then the model is reasonably simplified based on the complexity of the model and control performance. Currently, control strategies mainly include open-loop

**Funding:** This work was funded by the National Natural Science Foundation of China (Grant No. 52005344).

control, PID, visual feedback, and intelligent control [3–7]. Among them, the research on distributed hierarchical control theory for the soft octopus-like robot is typical. Pfeifer R et al. [8] from the University of Zurich used the distributed hierarchical control theory to interact the robot control system with the environment. Kuwabara J et al. [9] from the University of Tokyo transmitted video information collected by the visual system to the central nervous system, and then transmitted initial values and triggering signals to the peripheral nervous system to control the activity of arm muscles. Although the distributed hierarchical control system can guarantee a certain control accuracy, the system is complex and has high design costs. Guglielmino E et al. [10] from the Italian Institute of Technology added lower-level reflex control to the control system framework based on distributed hierarchical control theory to reduce the complexity of the distributed hierarchical open-loop control system. Cianchetti M et al. [11] optimized a distributed hierarchical control system to enable soft octopus-like robotic arms flexibly change posture in unstructured space. Han Z et al. [12] from the University of Science and Technology Beijing derived the dynamic model of the soft octopus-like robotic arm in the distributed parameter system based on the extended Hamilton principle and used Lyapunov to develop boundary control strategies to achieve distributed bending angle tracking. At present, distributed hierarchical control systems still require a large amount of communication and data transmission, which may have defects such as delay and data loss. Therefore, the distributed hierarchical control system also needs to improve data synchronization and increase the reliability of the system.

PID, visual feedback, and other closed-loop control strategies need further research on feedback from the internal and external environment of the robot to optimize the control model [13]. Katzschmann R K et al. [14] from the Massachusetts Institute of Technology first set the starting point, endpoint, and desired posture of the soft robot, and then used the PI, PID feedback curvatures, and visual feedback control algorithms to perform curvature closed-loop control on the robot, achieving end-path tracking. Wang H et al. [15] from Shanghai Jiao Tong University use visual servo control, shape control, and visual and force hybrid control algorithms to control soft robot to achieve the desired posture. Gong Z from Beihang University et al. [16] used visual feedback to obtain the relationship between air pressure and cavity elongation, achieving a certain position accuracy. Cosimo D S et al. [17] from the Massachusetts Institute of Technology developed a dynamic controller for soft robot based on closed-loop dynamic control theory and proposed a dynamic trajectory tracking and surface tracking architecture that can maintain the flexibility of the robot during motion. Hess A et al. [18] from Michigan State University optimized the parameters of the active strain amplitude and bias proportional integral, and differential controller to track different motion targets. This type of control algorithm has good stability and faster response speed, but it is difficult to combine with the structural characteristics and driving mode of the robot, resulting in low control accuracy and poor anti-interference. In addition, Best C M et al. [19] from Brigham Young University established a dynamic model based on the stiffness of pneumatic soft robots as the state variable to control the stiffness and posture transformation of the robot, and combined sliding mode and model predictive controllers to achieve precise control. Bruder D et al. [20] from Harvard University developed three model predictive controllers based on the Koopman system identification method and its application theory in model predictive controllers, and evaluated the performance of soft robotic arm control for multiple actual trajectory tracking tasks, verifying the accuracy of the control. Wu K et al. [21] from Lille University proposed a finite element-based soft robot trajectory tracking control strategy that can solve the control optimization problem in model prediction. However, these control methods only exist at the theoretical level, and cannot effectively verify that the control strategy can be well applied to soft robot system.

Because most of soft robot belong to bionic robots, it is difficult to consider the biological rhythm of bionic objects in the existing control strategy research, which makes the robot attitude transformation lack of flexibility, anti-interference, and interactivity [22–24]. In recent years, with the rapid development of robotics technology, many researchers have found that the CPG control strategy derived from the biological lower-level neural central control principle can form a closed-loop control with robots, which can quickly respond to external sensor feedback, couple output stable rhythmic control signals, and achieve real-time rhythmic motion control [25–27]. In the field of research on crawling bionic robots. Liljeback P et al. [28] from the Norwegian University of Science and Technology developed a control system for a pneumatic snake-like robot based on CPG, which can generate two different movement gaits: lateral undulation and side direction wriggling. The control parameters for the propulsion direction of the robot were also determined. In the research field of legged bionic robots. Mathias Thor et al. [29] from the University of Southern Denmark proposed a general motion control framework for legged robots based on CPG and Radial Basis Function Neural Network (RBFNN), which can be applied to three simulated legged robots with different morphologies. By optimizing the control strategy and integrating sensory feedback into the CPG-RBFNN network, online adaptation can be achieved. Daniel Gutierrez-Galan et al. [30] from the University of Seville proposed a CPG for hexapod robots, which can generate three types of motion gaits in real-time. CPG motion control also has certain stability and adaptability, and can play a good control effect in the process of robot jumping and running. Kassim A M et al. [31] from the University of Malaya developed a control system for a tripod jumping robot based on CPG, which can control parameters of each leg independently to enable the robot to perform jumps of different directions and heights. Kenichi Narioka et al. [32] from Osaka University developed an open-loop controller based on CPG, which uses controlled pneumatic artificial muscles to enable a quadruped robot to perform jumping movements. Generating sufficient torque in the legs allows the robot to kick the ground and contract its limbs, thus avoiding being tripped during movement. Takahiro Fukui et al. [33] from Ibaraki University studied the motion control of quadruped robots based on CPG, which can effectively enable the robot to complete stable walking and running respectively, and adjust their gaits autonomously to cross unknown obstacles. In addition, CPG motion control has been applied in multiple fields. Marlene H J et al. [34] from the University of Southern Denmark control the pneumatic three chambers soft robotic arm using a CPG and adapted it to external disturbances in real-time by using adaptive integral learning. Bhattacharya et al. [35] from the University of Auckland studied the motion control strategy of a soft swallowing robot based on CPG, which can produce different creeping trajectories to meet the job requirements. With the rapid development of CPG control algorithm research, it has gradually expanded to the field of underwater bionic robot motion control. Yu J et al. [36] from Peking University optimized the swimming parameters of multi-joint bionic robot fish based on CPG, and proposed a method to determine the relative optimization control parameters, which can improve the swimming performance of robotic fish. Wang Y et al. [37] from Northeast Forestry University proposed a bionic CPG pectoral fin waveform control strategy based on the Hopf oscillator, which enables robotic fish to achieve propulsion waveforms and swimming performance similar to those of biological stingrays. Fan Z et al. [38] from Nanjing University of Aeronautics and Astronautics drew on the swing pattern propulsion mechanism of fish and proposed a new propulsion mode that combines pectoral fin flapping and swinging based on CPG. The maximum average thrust is 2.8N and the maximum swimming velocity is 121mm/s. Li Z et al. [39] from Lanzhou Jiaotong University proposed a closed-loop motion control method based on CPG combined with fuzzy control, which can improve the dynamic performance and steady-state performance of depth control in robotic fish, increase the speed of robotic fish

approaching the desired depth, and reduce the steady-state error during depth cruising. Zhong Y et al. [40] from the South China University of Technology proposed a mechanical model that can describe different swimming postures of fish based on improved CPG, which can reduce the difficulty of multi-posture control of robotic fish and improve the control accuracy, and can also study the inherent characteristics of different movements of fish at low cost. Chen G et al. [41] from Zhejiang Sci-Tech University proposed a target tracking control system based on a bionic closed-loop CPG, which can dynamically adjust the robot motion in response to the data of visual sensors to achieve accurate target tracking, so that the bionic mantis shrimp robot can make multi-angle turns and flexibly adjust speed in a limited underwater space.

The CPG control strategy can autonomously generate a rhythmic motion with smooth transition trajectories even in the absence of external feedback signals, and has strong robustness and real-time accuracy, making it suitable for the research of control strategies for soft bionic robots [42, 43]. However, biological CPG is a multi-level complex neural network composed of multiple neurons, and existing CPG motion control research results often have shortcomings in integrity simulating of the biological prototypes, especially difficult to apply to the control requirements of bionic robots with relatively complex movements [44, 45]. This article improves the nonlinear neural oscillator based on the motion characteristics and driving methods of robot and combines the complementary characteristics of neural network and fuzzy theory to build a CPG-ANFIS-PID motion control system. It fully utilizes the learning ability of neural networks and the ability of fuzzy systems to solve fuzzy and qualitative knowledge, autonomously identifies fuzzy logic rules, sets membership functions, and corrects relevant parameters, achieving the accuracy and anti-interference of motion control for the pneumatic soft robot.

## 2 Establishment of improved CPG motion control model

The CPG motion control network is composed of bidirectional or unidirectional coupling of multiple neural oscillators, and each neural oscillator also contains excitation and inhibition neurons with mutual coupling and alternating inhibition characteristics. Among them, when a neuron is activated, it will change its current self-state to an excited state and inhibit another neuron connected to it, alternately generating self-excited oscillation signals, thus achieving rhythmic motion control [46].

### 2.1 Establishment of the Hopfield oscillator model

The pneumatic soft bionic robot mainly consists of bilateral flippers, a tail, and a head-neck structure as its executing mechanisms. To ensure the synchronicity and coordination of the robot control system and achieve the desired motion, multiple oscillators need to be mutually coupled. The Hopfield oscillator network topology control model is improved. A CPG network control model composed of 22 Hopfield oscillators is designed by the diffusion coupling method, whose frequency, amplitude, and phase can be adjusted separately, and the coupling term (i.e. perturbation vector) generated by the combination of the coupling matrix and the rotation matrix is added to the oscillators.

The expression for each CPG control model of the pneumatic amphibious soft bionic robot obtained is:

$$\begin{cases} \dot{x}_i = kx_i\left[m_i - x_i^2 - (y_i - b_i)^2\right] - 2\pi f_i(y_i - b_i) + h_1\left[x_{i-1}\cos\varphi_{i,i-1} + (y_{i-1} - b_{i-1})\sin\varphi_{i,i-1}\right] \\ \dot{y}_i = k(y_i - b_i)\left[m_i - x_i^2 - (y_i - b_i)^2\right] + 2\pi f_i x_i + h_2\left[x_{i+1}\sin\varphi_{i,i+1} + (y_{i+1} - b_{i+1})\cos\varphi_{i,i+1}\right] \end{cases} \quad (1)$$

Where $x$, and $y$ represent the state variables of oscillator excitation and inhibition neurons, $k$ is the convergence rate constant of the control oscillator, and $k > 0$; $\sqrt{m_i}$, and $f_i$ represent the amplitude and frequency of the $i$-th oscillator, respectively, $m > 0$, and $f > 0$, $h_1$, and $h_2$ is the weight coefficients, $\varphi_i$ represents the phase relationship between oscillators, $b_i$ is the offset.

It should be noted that when the offset value $b_i$ is changed, the CPG model is capable of generating asymmetric bias output signals, which can change the motion direction of the robot.

## 2.2 Improvement of the CPG oscillator model

For the control principle of the pneumatic amphibious soft bionic robot, its controller mainly inflates each executing mechanism of the robot by adjusting the pressure values through the electrical proportional valves. When the expected motion forms are achieved, the electromagnetic directional valve is used for deflating, while the electric proportional valve pauses inflation, i.e., the timing of pausing inflation of the electric proportional valve corresponds to that of deflating of the electromagnetic directional valve. The inflation and deflation time will be adjusted according to different motion forms of the executing mechanisms, depending on the motion amplitude, frequency, phase, and structural characteristics of the robot. Therefore, for the control system of the pneumatic soft bionic robot, it is necessary to improve the original Hopfield oscillator model to generate corresponding waveforms according to different motion modes.

The relationship between frequency $f$ and occupancy factor $\varepsilon$ is introduced into the CPG control model, which can be expressed as:

$$\begin{cases} f = \dfrac{f_{inflation}}{e^{-by_i} + 1} + \dfrac{f_{deflation}}{e^{by_i} + 1} \\ f_{deflation} = \dfrac{1 - \varepsilon}{\varepsilon} f_{inflation} \end{cases} \tag{2}$$

Where $f_{inflation}$ is the inflation phase frequency, $f_{deflation}$ is the deflation phase frequency, $b$ is a large normal number, which determines the conversion speed of $f$ between $f_{inflation}$ and $f_{deflation}$, $\varepsilon$ is the land occupation coefficient, which refers to the proportion of deflation phase in the whole moving process. When $\varepsilon = 1/2$, the inflation and deflation time are the same, the inflation and deflation time can be adjusted by changing the $\varepsilon$.

In addition, to enable soft bionic robot to switch modes by continuously changing parameters, it is necessary to make the phase difference and land occupation coefficient continuously change from the corresponding value of the current mode to the corresponding value of the target mode within a certain time range. The expression for the change process is as follows:

$$\begin{cases} \varphi_{ij}^+ = \varphi_{ij}^+ + \dfrac{\varphi_{ij}^- - \varphi_{ij}^+}{e^{\kappa(t-t_0)}} \\ \varepsilon^+ = \varepsilon^+ + \dfrac{\varepsilon^- - \varepsilon^+}{e^{\kappa(t-t_0)}} \end{cases} \tag{3}$$

Where $\varphi_{ij}^+$ and $\varepsilon^+$ are the phase difference and land occupation coefficient of the target mode, $\varphi_{ij}^-$ and $\varepsilon^-$ are the phase difference and land occupation coefficient of the actual mode, $t_0$ is the time of receiving the mode change command, and $t$ is the accumulated time since time $t_0$, $\kappa$ is used to adjust the speed of parameter change and determine the stability of mode switching, the greater the value $\kappa$, the faster the speed of change, $\kappa = 1.2$ is taken.

## 2.3 CPG control parameters adjustment

Adjusting CPG control parameters mainly involves input parameters of amplitude, frequency, phase difference, offset, and coupling coefficient. Input parameters of frequency, phase difference, and offset can be adjusted based on the characteristics of the robot structure, actual motion situation, and relevant experience. Here, the focus is on exploring the adjusting of input parameters for coupling coefficient and amplitude.

Among them, for adjusting the input value of the coupling coefficient, it is necessary to first determine the propulsion coupling relationships between the same type of mechanisms and different types of mechanisms, and then determine the robot phase coupling relationship (the phase lag relationship between executing mechanisms), the ratio of motion frequencies, and the ratio of motion amplitudes. It should be noted that since pneumatic amphibious soft bionic robot are driven by various actuators, even the same executing mechanism is driven by independent actuators. Therefore, the coupling relationship of the soft bionic robot needs to be determined based on the coupling relationship between different types of propulsion components. For the determination of the coupling coefficient between the flipper and the actuator, it is assumed that the maximum angular velocity of the actuator driving the flipper on one side is $\omega_{fmax}$ under normal working conditions, and the maximum angular velocity of the actuator driving the flipper on the other side is $\omega_{fmax}$, and the ratio of the two is taken as its coupling coefficient, that is $\omega_{fmax}/\omega_{jmax}$. The coupling coefficient between the tail and the actuator is determined by the tail angle of the attack method. To ensure a better propulsion effect, based on the theoretical basis given by Lighthill, the Expression (4) of the maintaining feathering parameter is obtained as follows:

$$\rho = \alpha_{\max} v / \omega_{\max} \tag{4}$$

Where $\alpha_{\max}$ is the maximum angle of attack, $\omega_{\max}$ is the maximum value of lateral swing angular velocity, $v$ is the velocity of movement.

When $\rho = 0.6 \sim 0.8$, better propulsion efficiency can be obtained. Therefore, the coupling coefficient between the tail and each actuator can be determined according to the angle of attack $\alpha_{\max}$. In addition, the principle of determining the coupling coefficient between the head-neck and the actuator is the same as that of the tail.

For the adjusting of the amplitude input parameter, as the soft bionic robot is a nonlinear system, it is necessary to define the amplitude input end based on the mechanical models of the various actuators of the robot. It should be noted that the mechanical model of the robot has a certain complexity, and it is difficult to directly incorporate it into the amplitude input end of the CPG control module. Therefore, this article collects simulation results of the mechanical model and obtains the empirical formulae of each actuator mechanical model through fitting algorithms, and places the empirical formulae into the amplitude input end of the CPG control module. When different numerical pressures are input, corresponding motion amplitudes can be obtained. Among them, it can be known from the fitting that the empirical formula is expressed by a one-variable quartic equation, and the fitting rates can reach more than 98%, which meets the actual control requirements. The specific form of the empirical formula is as follows:

$$f(x) = p_1 x^4 + p_2 x^3 + p_3 x^2 + p_4 x + p_5 \tag{5}$$

Where $p_1$, $p_2$, $p_3$, $p_4$ and $p_5$ are empirical expression coefficients, which are constants, $x$ is the inflation pressure, $f(x)$ is the motion angle.

## 2.4 Establishment of CPG topology structure

The pneumatic amphibious soft bionic robot is a relatively complex high-dimensional and strongly coupled nonlinear system, which requires flexible, smooth, and stable coordinated motion characteristics for its various executing mechanisms. Based on this, a two-layer CPG motion controller is proposed in this article, where the first layer is the consciousness layer consisting of 4 oscillators with bidirectional coupling, which can each correspond to the left and right flippers, tail, and head-neck region of the robot, and are connected via fully symmetric mesh topology. The second layer is the behavior layer consisting of 18 oscillators, each bidirectionally coupled with the first layer, used to realize the interaction between the four actuators of the robot and their respective degrees of freedom, connected via a multi-chain bidirectional coupling structure. The consciousness layer determines the a priori knowledge of the robot before performing the motion, while the behavior layer determines the ability of the robot to perform the motion, playing a critical role in determining the final motion state and performance.

It should be noted that to reduce network coupling and computational costs, the nodes in the second layer network model are only directly bi-directionally coupled to the nodes in the first layer network, and there is no coupling relationship between the other nodes in the second layer. Control requirements are achieved by stabilizing the phase difference locked between the first-layer nodes and adjacent second-layer nodes. The specific process is to use the first layer nodes to transfer the expected locked control signal phase difference to the internal state variable.

$x_i$ of the second layer node based on the external disturbance vector, and dynamically adjust the state variable $y_i$ to its corresponding first layer nodes state variable until the expected phase difference can be stably locked, achieving stable coupling signal output with a phase difference. Therefore, in robot motion mode switching, the control model can be quickly reconstructed in real-time with the changes in the first layer nodes and has the characteristics of high sensitivity, low computational cost, and easy implementation. In addition, the output state signals of each oscillator in the CPG network correspond to the controlled freedom degrees of each actuator, which allows form waveforms of various motion modes to be formed by the actuators. The CPG structure topology of the pneumatic soft bionic robot is shown in Fig 1.

## 2.5 Mapping function establishment

Since the output signal of the CPG model oscillator is dimensionless, it is necessary to use a mapping function to convert the output signal into a robot motion control signal, and to adjust the coupling input between the executing mechanisms of the robot to achieve the required motion waveforms for each motion mode. Based on the driving characteristics of the pneumatic system, adjust the output amplification function of the motion freedom degrees of each actuator to return to 0 during deflation and maintain it until the end of the entire deflation stroke. For the output signal $x_i$ acting on the electrical proportional valve, the modified output amplification function is expressed by the following Eq (6):

$$z_i = \begin{cases} c_i x_i + b_i, & (0 < x_i \leq x_{i\max}) \\ c_i x_{i\max} + b_i, & (x_i > x_{i\max}) \\ 0, & (x_i = 0) \end{cases} \tag{6}$$

Where $z_i$ is the $i$-th oscillator output processed by the system, $x_i$ is the output signal of the $i$-th CPG, and $x_{i\max}$ is the output limiting threshold of the $i$-th oscillator; $c_i$ is the output signal amplification coefficient; $b_i$ is the output offset of the $i$-th oscillator.

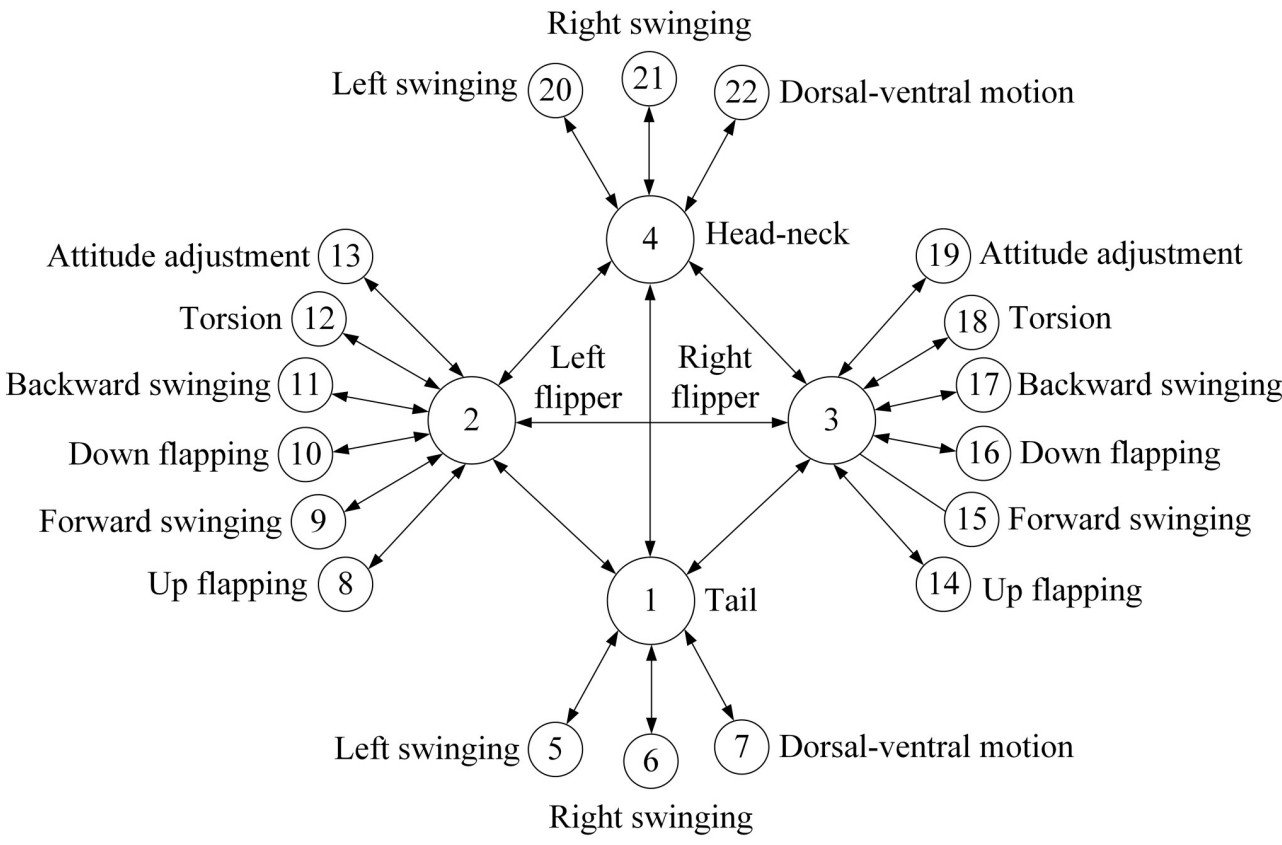

**Fig 1. Topological structure diagram of the CPG network of the robot.**

## 3 Construction and simulation of CPG-ANFIS-PID motion control strategy

Although the CPG motion control system can achieve certain control requirements, the pneumatic system and silicone rubber material of soft robot can seriously affect the control accuracy. Therefore, an adaptive, self-learning, and self-adjusting control strategy is needed to compensate for the errors in the nonlinear motion control system. This article combines the complementary characteristics of neural networks and fuzzy theory to build an ANFIS, which can fully utilize the learning ability of neural networks and the ability of fuzzy systems to deal with fuzzy and qualitative knowledge. In addition, the ANFIS can autonomously identify fuzzy logical rules, adjust the membership functions, and modify relevant parameters. Finally, the CPG-ANFIS-PID motion control system is validated through a simulation platform.

### 3.1 ANFIS structure and learning algorithm

ANFIS belongs to a local approximation network, which mainly embeds neural network learning algorithms within the fuzzy control system, and completes fuzzification, fuzzy inference, and de-fuzzification respectively through the self-learning and offline training capabilities of the neural network. Among them, based on sufficient empirical knowledge, the nonlinear mapping of inputs and outputs can be effectively approximated infinitely using neural network learning algorithms [47, 48].

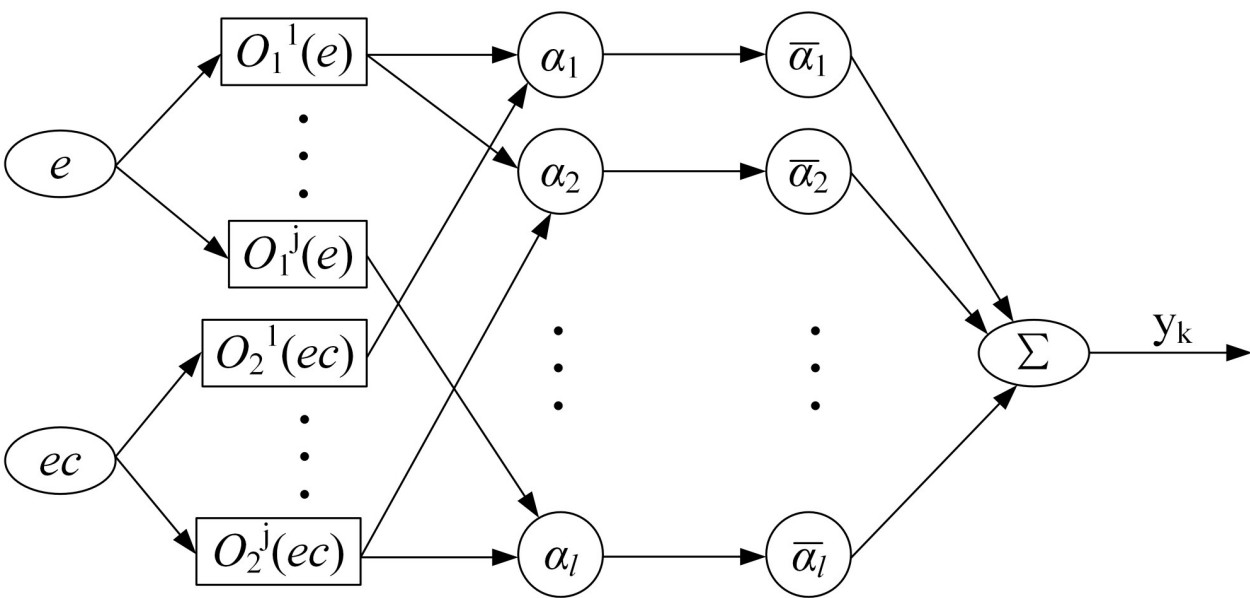

First layer  Second layer   Third layer   Fourth layer   Fifth floor

**Fig 2. ANFIS model structure.**

(1) Structure of ANFIS

The ANFIS structure consists of the input layer, the membership function generation layer, the inference layer, the normalization layer, and the de-fuzzification layer, as shown in Fig 2.

(2) The learning algorithm of ANFIS

When determining the input variables and fuzzy language variables of ANFIS, for multi-layer feed-forward network structure, only the central value $c_{ij}$, width $\sigma_{ij}$, and connection weight $w_{ij}$ of the membership function need to be studied and modified. Among them, the mathematical expressions of nodes in each layer are respectively represented by the following forms.

The Expression (7) for the input layer:

$$\begin{cases} f_i^{(1)} = x_i \\ O_{ij}^{(1)} = f_i^{(1)} \end{cases} \tag{7}$$

Where $i = 1, 2,\dots,n, j = 1,2,\dots,m_i$

The Expression (8) for the membership function layer:

$$\begin{cases} f_{ij}^{(2)} = -\dfrac{\left(x_i - c_{ij}\right)^2}{\sigma_{ij}^2} \\ O_{ij}^{(2)} = \exp\left(f_{ij}^{(2)}\right) = O_i^j \end{cases} \tag{8}$$

The Expression (9) for the fuzzy inference layer:

$$\begin{cases} f_l^{(3)} = min\{O_1^{i_1} \cdot O_2^{i_2} \cdot \cdots \cdot O_n^{i_n}\} \text{或} f_l^{(3)} = O_1^{i_1} \cdot O_2^{i_2} \cdot \cdots \cdot O_n^{i_n} \\ O_l^{(3)} = \alpha_l = f_l^{(3)} \end{cases} \tag{9}$$

Where $l = 1,2\ldots,49$

The Expression (10) for the normalization layer:

$$\begin{cases} f_l^{(4)} = O_l^{(3)} / \sum_{i=1}^{m} O_i^{(3)} = \alpha_l / \sum_{i=1}^{m} \alpha_i \\ O_l^{(4)} = \bar{a}_l = f_l^{(4)} \end{cases} \tag{10}$$

The Expression (11) for the inference result output layer:

$$\begin{cases} f_k^{(5)} = \sum_{k=1}^{m} w_{ij} \cdot O_l^{(4)} = \sum_{k=1}^{m} w_{ij} \cdot \bar{\alpha}_l \\ O_k^{(5)} = y_k = f_k^{(5)} \end{cases} \tag{11}$$

In addition, using an incremental PID control algorithm, Expression (12) is:

$$u(k) = u(k-1) + \Delta u(k) \tag{12}$$

Where $u(k)$ is the controller output.

$$\Delta u(k) = f_k^{(5)} \cdot xc = k_p[e(k) - e(k-1)] + k_i e(k) + k_d[e(k) - 2e(k-1) + e(k-2)] \tag{13}$$

Where $k_p = f_1^{(5)}, k_i = f_2^{(5)}, k_d = f_3^{(5)}$

In the ANFIS-PID controller, the network connection weights $w_{ij}$, and the center values $c_{ij}$ and width values $\sigma_{ij}$ of the membership functions of the fuzzy layer neuron nodes need to be learned separately. Back-propagation method is used for parameters adjustment, and the mathematical Expression (14) for the error is taken as:

$$e = \frac{1}{2} \sum_{k=1}^{r} (d_k - y_k)^2 \tag{14}$$

Where $d_k$ is the expected output gain, $y_k$ is the actual output gain.

According to Expression (15), the error control interval for each iteration step is $d_k - y_k$. To make the system output $y_k$ statistically closest to the expected output $d_k$, it is necessary to minimize $e$. Here, according to the gradient descent method, the error function is minimized by continuously adjusting the connection weight value $w_{ij}$.

Finally, the learning method for parameter correction can be expressed as:

$$\begin{cases} w_{ij}(t+1) = w_{ij}(t) - \eta \dfrac{\partial e}{\partial w_{ij}} + \alpha\Big[w_{ij}(t) - w_{ij}(t-1)\Big] i = 1, 2, \ldots, n; j = 1, 2, \ldots, m_i \\[2mm] c_{ij}(t+1) = c_{ij}(t) - \eta \dfrac{\partial e}{\partial c_{ij}} + \alpha\Big[c_{ij}(t) - c_{ij}(t-1)\Big] i = 1, 2, \ldots, n; j = 1, 2, \ldots, m_i \\[2mm] \sigma_{ij}(t+1) = \sigma_{ij}(t) - \eta \dfrac{\partial e}{\partial \sigma_{ij}} + \alpha\Big[\sigma_{ij}(t) - \sigma_{ij}(t-1)\Big] i = 1, 2, \ldots, n; j = 1, 2, \ldots, m_i \end{cases} \tag{15}$$

Where $t$ is the network iteration algebra. $\eta$ is the learning rate, taking a positive value. $\alpha$ is the inertia coefficient, $0 < \alpha < 1$.

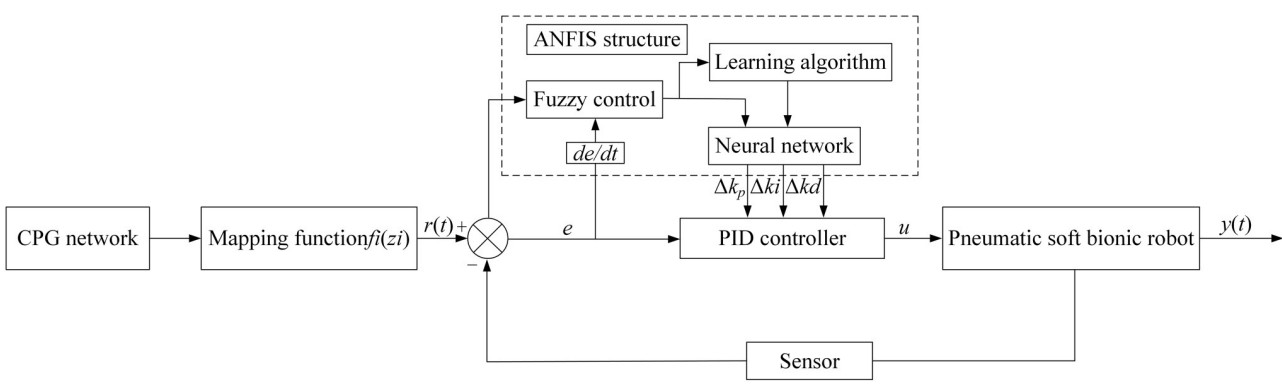

**Fig 3. CPG-ANFIS-PID control principle diagram.**

## 3.2 Construction of CPG-ANFIS-PID motion control strategy

ANFIS is a learning method that extracts corresponding fuzzy rules from a dataset. The internal mechanism combines the Back-propagation algorithm with the least squares method. Based on the obtained hybrid algorithm and using input and output data as objects to establish a model, it is possible to obtain the optimal parameters for the membership function, so that the expected input and output relationship can be more effectively simulated through a Takagi-Sugeno type fuzzy inference system. The principle of CPG-ANFIS-PID motion control is shown in Fig 3.

It can be seen from Fig 3 that the control system mainly includes a CPG module, hybrid learning algorithm, fuzzification module, and PID controller, which can continuously detect $e$, and $ec$ according to the fuzzy relationship between error $e$ and error change rate $ec = de/dt$, and dynamically adjust $k_p$, $k_i$, $k_d$ in real-time online through fuzzy inference. It can ensure the control parameters requirements of $e$, and $ec$ at any time, and make the soft bionic robot system have superior control performance, to realize the operation function in different environments. Since ANFIS can only build single-output Takagi-Sugeno type, this article designs 18 parallel two-input single-output ANFIS-PID structures, sets fuzzy state according to the error $e$ and error change rate $ec$ between the theoretical amplitude and the actual amplitude reference value, and dynamically adjusts the correction parameters $\Delta k_p$, $\Delta k_i$, and $\Delta k_d$. Among them, the Gauss function and constant are defined into the membership function types of input and output variables respectively. For the fuzzy control rule table corresponding to its modified parameters $\Delta k_p$, $\Delta k_i$, and $\Delta k_d$, the grid division method is used for training, and the structural parameters in the fuzzy inference system are optimized and adjusted.

## 3.3 Simulation research on CPG-ANFIS-PID motion control

Based on the constructed CPG-ANFIS-PID motion control model, the motion control performance of a pneumatic soft bionic robot is simulated and studied.

(1) Research on the simulation of the soft bionic robot in the straight swimming mode

Robots mainly achieve straight swimming through the coordinated movement of flippers and tail. Among them, both sides of the flippers movement in the same frequency, amplitude, phase difference, and initial position offset is 0 for flapping and swinging and positional rotation motion. At the same time, the tail can move in two modes: left-right swinging and dorsal-ventral mode, and this straight swimming mode is used for simulation. At slow, moderate, and

high velocities, the frequencies of each oscillator are 0.6Hz, 0.8Hz, and 1Hz, and the amplitudes of the flippers flapping and swinging oscillator are 0.52 rad, 0.79 rad, and 1.05 rad. The amplitudes of the positional rotation oscillator are 0.52 rad, 0.70 rad, and 0.87 rad, with coupling coefficients are 1.5. The amplitudes of the tail left-right swinging oscillators are 0.52 rad, 0.79 rad, and 1.05 rad, and the amplitudes of the dorsal-ventral motion oscillator are 0.87 rad, 1.22 rad, and 1.57 rad, with coupling coefficients are 1. Obtain the CPG output results of various modes of robot straight swimming, as shown in Fig 4.

According to Fig 4, it can be seen that the output signals of the CPG output signal can quickly and stably converge to the desired trajectory after a short transient under different parameter conditions. Among them, neurons 1 to 4 mainly transmit control signals to various executing mechanisms through the output frequency and phase. Neurons 5 to 7 control tail left swinging, right swinging, and dorsal-ventral movement through the output amplitude, frequency, and phase difference. Similarly, neurons 20 to 22 control the head-neck. Neurons 8 to 13 control the upward swinging, forward swinging, downward swinging, backward swinging, torsional, and posture adjustment of the left flipper through the output amplitude, frequency, and phase. Similarly, neurons 14 to 19 control the right flipper. It should be noted that the frequency and phase of the autonomous output by the robot can match the input parameters of the controller, and the output amplitude has a certain error with the input parameters of the controller amplitude. However, the error is small, which still indicates the feasibility of improving CPG for motion control. In addition, even though the amplitude and frequency of the control parameters are different during various mode switching, the transition trajectory of the output signal can still be maintained smoothly and flexibly.

(2) Research on the simulation of the soft bionic robot in the turning bow mode

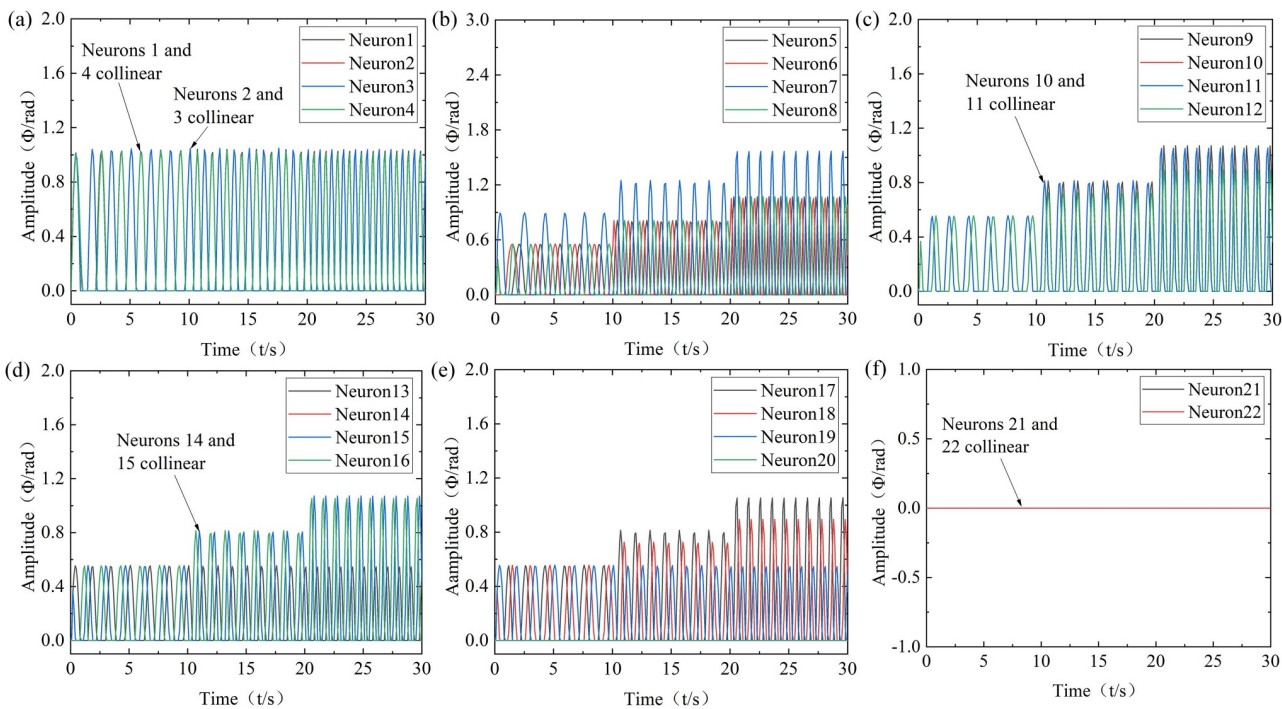

**Fig 4. Robot straight swimming modes.** t = 0~10s: Slow-velocity; t = 10~20s: Moderate-velocity; High-velocity: t = 20~30s. (a) Neurons 1 to 4, (b) Neurons 5 to 8, (c) Neurons 9 to 12, (d) Neurons 13 to 16, (e) Neurons 17 to 20, (f) Neurons 21 to 22.

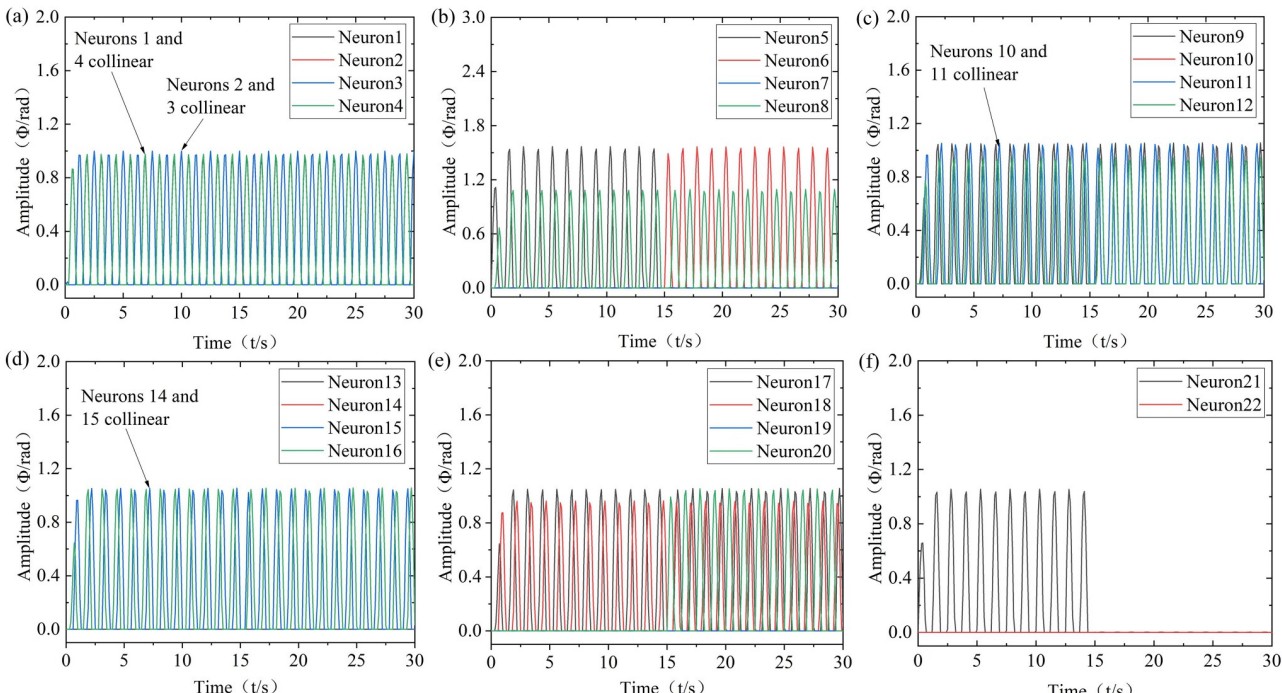

**Fig 5. Robot bow turning modes.** t = 0~15s: left turn bow; t = 15~30s: right turn bow. (a) Neurons 1 to 4, (b) Neurons 5 to 8, (c) Neurons 9 to 12, (d) Neurons 13 to 16, (e) Neurons 17 to 20, (f) Neurons 21 to 22.

The robot mainly achieves turning bow through the coordinated movement of flippers, tail, and head-neck. Among them, both sides of the flippers perform differential motion with the same frequency, same amplitude, opposite phase difference, and offset is 0 in different directions. At the same time, the tail swings in a "C" shape and adds an offset for collaborative cooperation, and the head-neck is assisted in collaboration to simulate the turning bow mode. During the process of turning bow to the left, the frequency of each oscillator is 0.8 Hz, the amplitude of the flipper flapping and swinging oscillators is 1.05 rad, the amplitude of the positional rotation oscillator is 0.87 rad, and the coupling coefficients are all 1.5. The amplitude of the tail left-right swinging oscillators is 1.57 rad, and the coupling coefficients are all 1. During the process of turning bow to the right, the differential direction of the flippers is opposite, and the swing and offset of the tail, and head-neck are also opposite to the turning bow to the left, with the other settings being the same. Obtain the CPG output results of each mode of robot turning bow, as shown in Fig 5.

According to Fig 5, it can be seen that the output signals of each oscillator of the robot can autonomously stabilize the desired trajectory of the controller. The output amplitude and frequency of the flipper swing oscillators for left-right turning bows are the same, and the phase remains constant. When transitioning from turning left to turning right, although the configuration of the CPG coupling phase difference changes, the waveform still maintains a smooth and flexible transition trajectory.

(3) Research on the simulation of the soft bionic robot in the snorkeling mode

The robot continuously adjusts the angle of attack of the wing fins through the positional rotation of the flippers and coordinates with the tail to achieve snorkeling. Among them, both sides of the flippers perform flapping and swinging movements with the same frequency,

amplitude, and phase difference. At the same time, the tail moves in left-right swinging or in a dorsal-ventral movement to simulate the snorkeling mode. During the upward floating process, the frequency of each oscillator is 0.8Hz, the amplitude of the flipper flapping and swinging oscillator is 1.05 rad, the amplitude of the positional rotation oscillator is 0.87 rad, the left flipper positional rotation offset is 4.5, the right flipper positional rotation offset is -4.5, and the tail amplitude is the same as that of high-velocity straight swimming. During the diving process, the left flipper positional rotation offset is -4.5, the right flipper positional rotation offset is 4.5, and the other parameter settings are the same as upward floating. Obtain the CPG output results of various modes of robot snorkeling, as shown in Fig 6.

According to Fig 6, it can be seen that the output amplitude and frequency of the flipper of the robot positional rotation oscillators are the same, the offset direction is opposite, and it can correspond to the input parameters of the controller. Moreover, the transition trajectory between the snorkeling modes is smooth and flexible.

(4) Research on the simulation of the soft bionic robot in the switching swimming mode

During the switch of swimming mode, the straight mode is set to moderate velocity, turning mode is set to left turn, and snorkeling mode is set to diving. The CPG-ANFIS-PID output results for the robot swimming mode switching are shown in Fig 7.

According to Fig 7, the CPG-ANFIS-PID control strategy can effectively compensate for the errors in the output amplitudes of the oscillators of the robot, which is more consistent with the input parameters of the controller. The output frequency and phase are still accurate, and there is no sudden change in mode switching. Therefore, by switching the swimming mode of the robot, it can be demonstrated that the CPG-ANFIS-PID controller can achieve the expected control effect and improve the control accuracy of the robot.

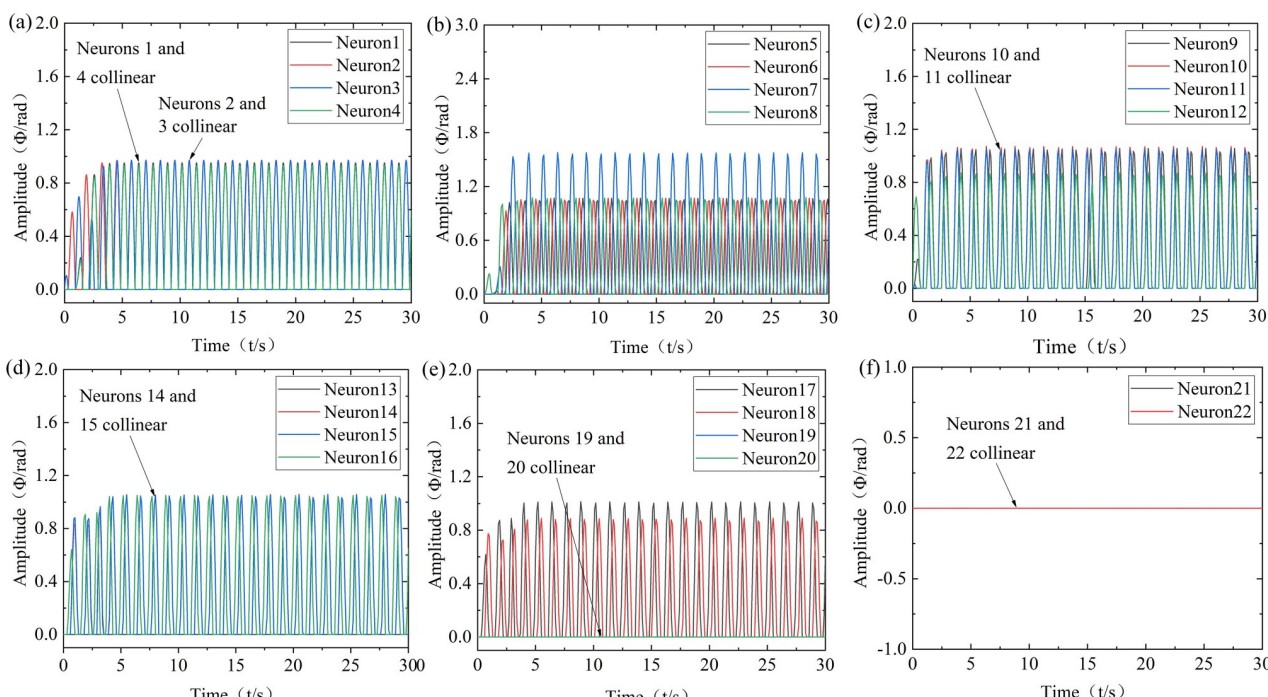

**Fig 6. Robot snorkeling modes.** t = 0~15s: Floating; t = 15~30s: Diving. (a) Neurons 1 to 4, (b) Neurons 5 to 8, (c) Neurons 9 to 12, (d) Neurons 13 to 16, (e) Neurons 17 to 20, (f) Neurons 21 to 22.

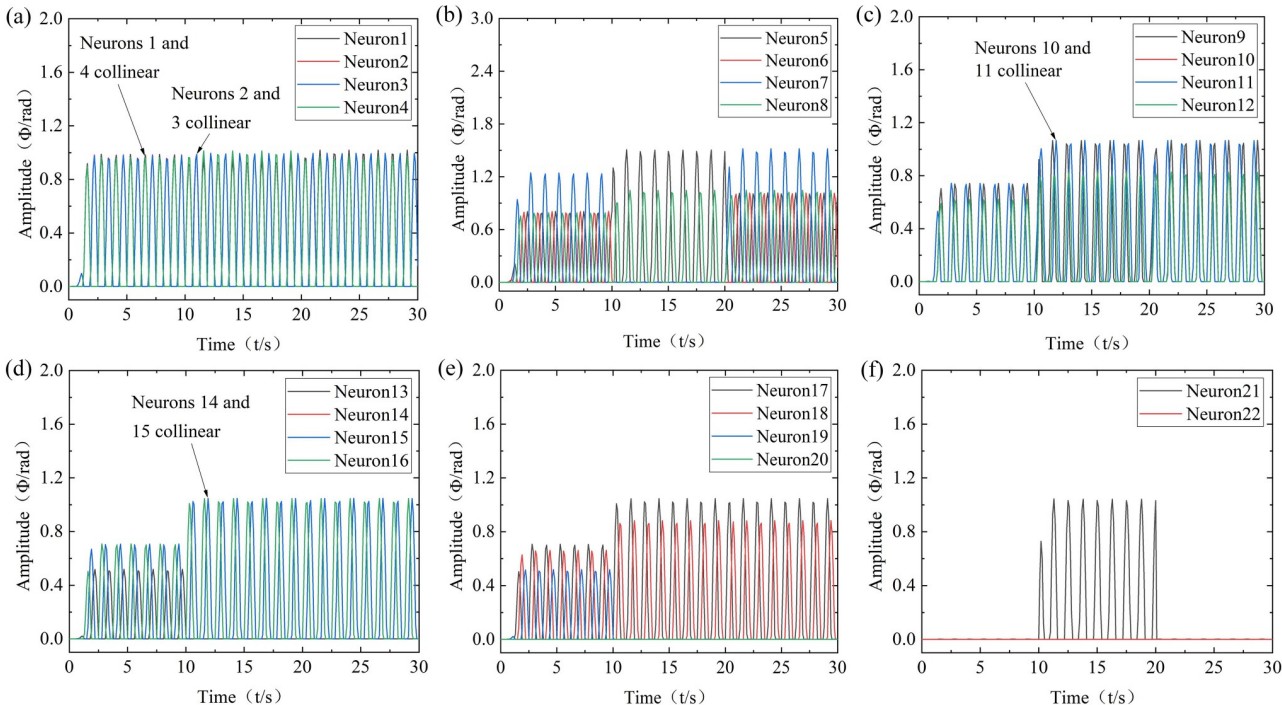

**Fig 7. Modes switching.** t = 0~10s: Moderate-velocity; t = 10~20s: Left turn bow; t = 20~30s: Diving. (a) Neurons 1 to 4, (b) Neurons 5 to 8, (c) Neurons 9 to 12, (d) Neurons 13 to 16, (e) Neurons 17 to 20, (f) Neurons 21 to 22.

In order to further verify the advantages of the CPG-ANFIS-PID control algorithm, the simulation results of CPG-ANFIS-PID and CPG motion control were compared. Taking the modal transformation of low-velocity and medium-velocity swimming of robot as an example, the data of typical neurons of 5, 7, 9, 10, 14, 17, 18, and 21 were compared and analyzed respectively, as shown in Fig 8.

According to Fig 8, compared to the CPG motion control strategy, it can be further verified that the CPG-ANFIS-PID control strategy can better match the input parameters of the controller, achieve better error compensation for the output amplitudes of each oscillator of the robot, and have good control effects.

## 4 Experimental research

### 4.1 Construction of control system

The control system of the pneumatic soft bionic robot adopts a hierarchical layout, mainly including the organizational layer, coordination layer, and execution layer. Among them, the organizational layer is the high-level center of organisms, which corresponds to high-level motion planning. With PC-104 as the upper computer, it can send control commands to the coordination layer. The coordination layer is the low-level center of organisms, corresponding to the CPG-ANFIS-PID controller, which controls the low-level motion. Using STM32F103C8T6 as the motion controller, it can receive high-level center commands and low-level center feedback, and send motion control commands to the pneumatic soft bionic robot actuator of the low-level [49]. The specific implementation process is based on the CPG-ANFIS-PID motion control algorithm, which outputs a PWM signal through pins. The conversion module converts the PWM signal into a voltage value to control the pressure of the

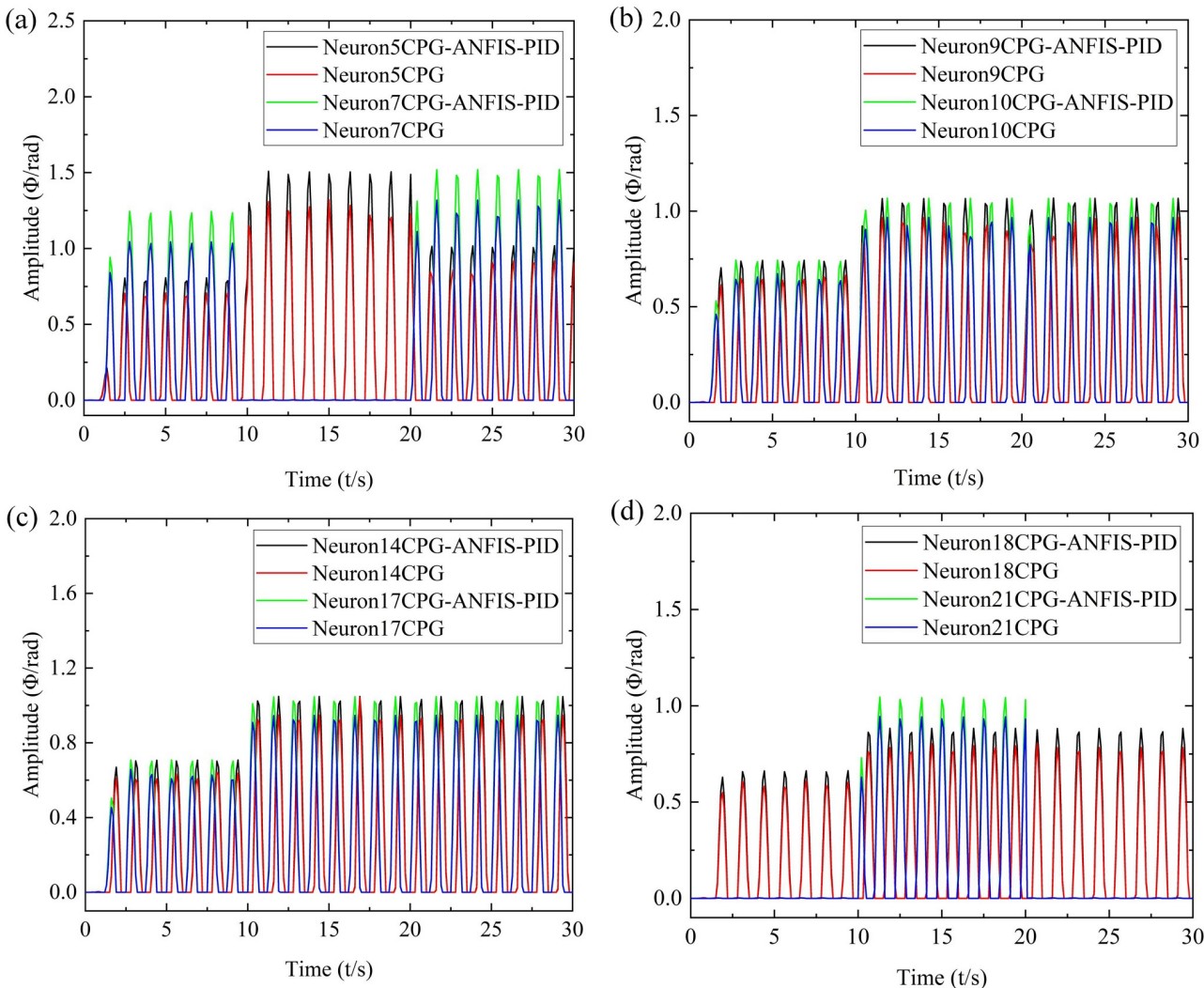

**Fig 8. Comparison of simulation results between CPG-ANFIS-PID and CPG.** (a) Neurons 5 and 7, (b) Neurons 9 and 10, (c) Neurons 14 and 17, (d) Neurons 18 and 21.

electrical proportional valve, and then sends commands through the relay to control the on/off timing of the electromagnetic directional valve in the system, generating corresponding motion control signals. The execution layer is the effector in the biological motion control system, composed of a pneumatic system and a soft bionic robot actuator. Obtain the principle of the robot control system, as shown in Fig 9.

## 4.2 Experimental testing analysis

**4.2.1 CPG-ANFIS-PID motion control algorithm validation.** Taking the low-velocity and medium-velocity swimming of pneumatic soft bionic robot as examples, control the coordinated movement of the two sides of the flippers, tail, and head-neck of the robot, test the motion performance of each executing mechanism, and verify the CPG-ANFIS-PID motion control algorithm. In the experiment, a camera was used to record the motion time and trajectory of the robot, and the swimming distance was measured using marker points and rulers. In

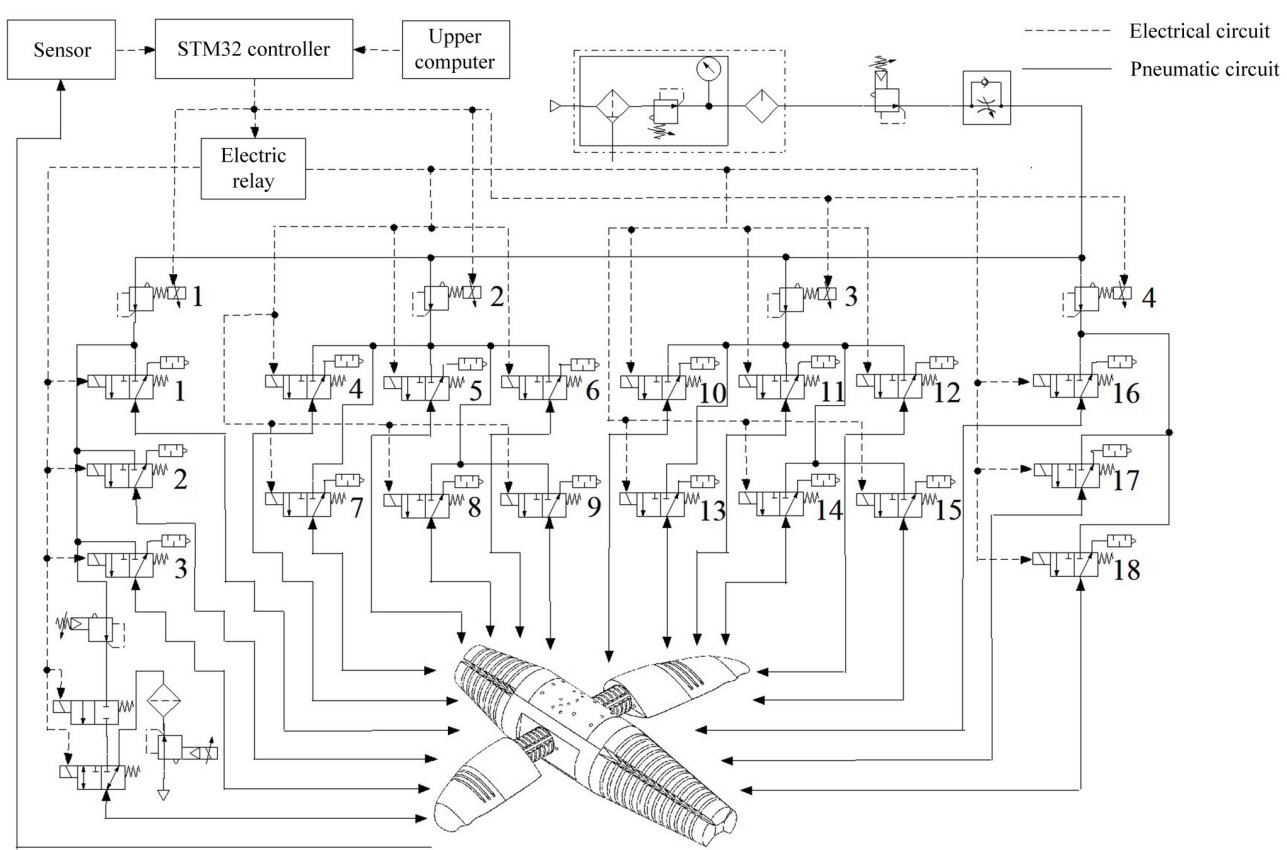

**Fig 9. Schematic diagram of the control system.**

addition, during the testing process of the robot, the specific input parameters of each oscillator are set the same as those in the simulation. The robot testing process is shown in Fig 10.

Collect typical neurons during the process of modal transformation in robot swimming, and collect data from neurons 5, 7, 9, 10, 14, 17, 18, and 21, and compare the data with the corresponding CPG-ANFIS-PID and CPG control results, as shown in Fig 11.

According to Fig 11, it can be seen that the experimental results of the pneumatic soft bionic robot under the control of the CPG-ANFIS-PID controller are slightly smaller than the simulation results, but the overall curve distribution trend is consistent with the simulation result. Therefore, it can be confirmed that the CPG-ANFIS-PID control algorithm is suitable for the motion control of the pneumatic soft bionic robot. The reason for the experimental test data being smaller than the simulation result is mainly due to factors such as the pneumatic system, underwater buoyancy, and human factors during the underwater experiment process of the robot. In addition, although the overall curve distribution trend of the CPG control results is consistent with the CPG-ANFIS-PID control results, the lower CPG output results can further verify the superiority of the CPG-ANFIS-PID control algorithm.

**4.2.2 Swimming performance test.** According to CPG-ANFIS-PID motion control strategy, through the coordinated motion of flippers, tail, and head-neck of pneumatic soft bionic robot, the straight swimming, turning and snorkeling movements are completed respectively. The swimming state of the robot is shown in Fig 12.

(1) The straight swimming of the robot performance test.

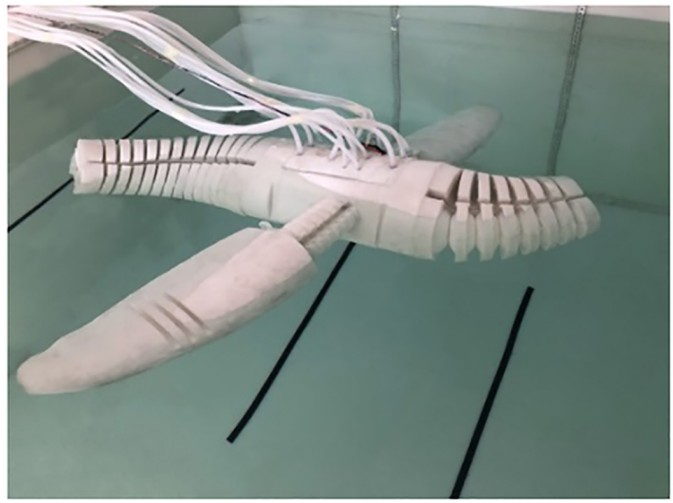

**Fig 10. Straight swimming motion of coordinated motion of each executing mechanism of the robot.**

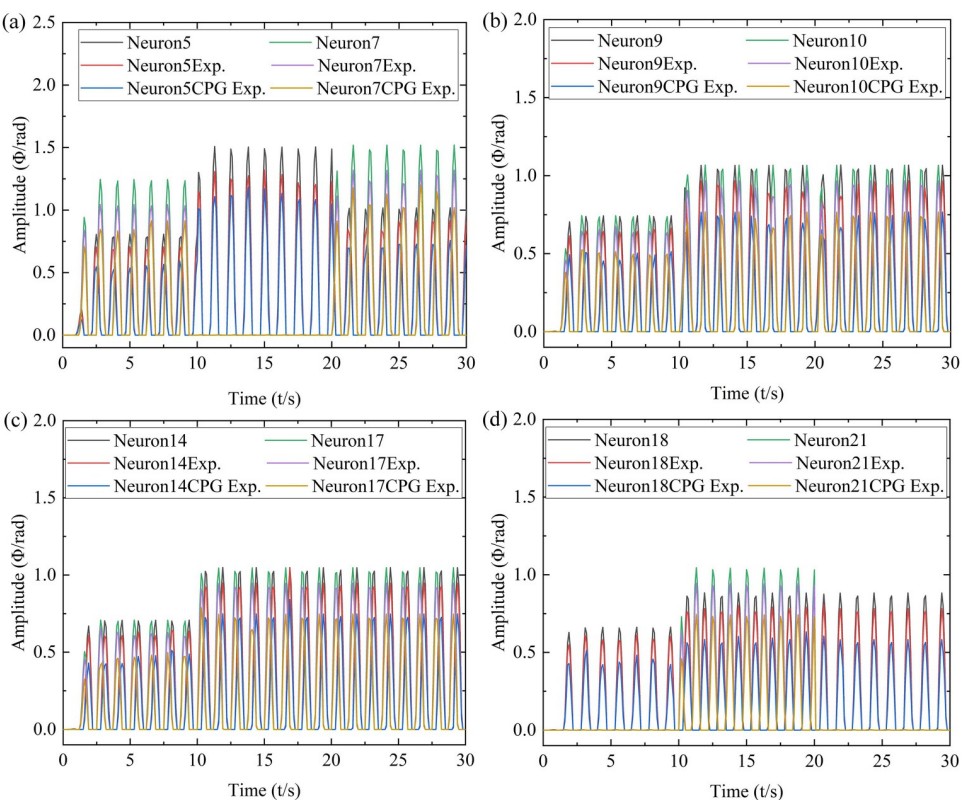

**Fig 11. Straight swimming motion of coordinated motion of each actuator of the robot.** (a) Neurons 5 and 7, (b) Neurons 9 and 10, (c) Neurons 14 and 17, (d) Neurons 18 and 21.

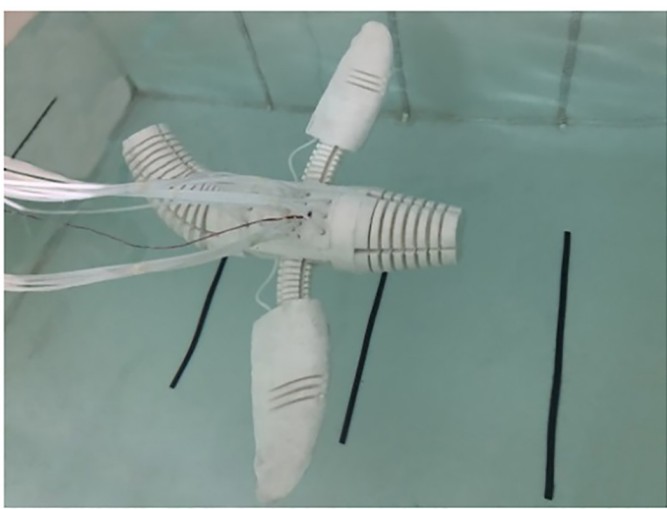

**Fig 12. Swimming performance test of soft bionic robot.**

**Table 1. Test results of cooperative motion of each actuator of the soft bionic robot in straight swimming.**

| Straight swimming form | Distance $s$(mm) | Time $t$(s) | Velocity $v$(mm/s) |
|---|---|---|---|
| Flippers+tail left-right swinging | 10000 | 53.9 | 185.5 |
| Flippers+tail dorsal-ventral movement | 10000 | 58.6 | 170.6 |

The straight swimming of the soft bionic robot mainly includes the simultaneous, same frequency and same direction flapping and swinging of the flippers on both sides+left-right swinging of the tail, and the simultaneous, same frequency and same direction flapping and swinging of the flippers on both sides + dorsal-ventral movement of the tail. The motion frequency of flippers on both sides of the robot is 1Hz, the amplitude is 0.87rad, the tail motion frequency is 0.8Hz, and the amplitude of the left-right swing and dorsal-ventral swing is 0.52 and 0.78rad respectively. It should be noted that when the tail motion frequency resonates with the intrinsic frequency of the flipper, the swimming velocity of the tail-driven robot can reach the peak. Therefore, to cooperate with the flippers for efficient coordinated movement, the tail movement frequency is lower than that of the flippers. Collect and process the test data of the straight swimming experiment of the robot, and the specific results are shown in Table 1.

According to Table 1, the swimming velocities of the two straight swimming modes of the robot are 185.5mm/s and 170.6mm/s, respectively.

(2) Performance testing of the robot turning bow

The turning bow motion modes of the pneumatic amphibious soft bionic robot mainly include differential (i.e. asynchronous cooperative motion) of the flippers on both sides and the "C" shape swinging of the tail, and the head-neck can also play an auxiliary role in turning bow.

During the process of the robot turning bow to the left, the motion frequency of the right flipper is 1Hz, and the motion amplitude is 0.78 rad. The motion direction of the left flipper is opposite to that of the right flipper, and the parameter settings are the same; The frequency of

**Table 2. Test results of the soft bionic robot in turning bow.**

| Turning bow form | Radius $r$(mm) | Time $t$(s) | Velocity $v$(rad/s) |
|---|---|---|---|
| Turn to the left | 772 | 32 | 0.21 |
| Turn to the right | 785 | 32 | 0.22 |

the tail swinging to the right is 0.8Hz, and the swing amplitude is 1.57rad. For the right-turning bow of the robot, the motion direction of flippers, tail, and head-neck on both sides is opposite to that of turning bow to the left, while the other parameter settings are the same. Collect and process the test data of the robot turning experiment separately, and the specific results are shown in Table 2.

According to Table 2, the left and right turning bow velocities of the robot are 0.21 rad/s and 0.22 rad/s, respectively. Among them, there is a certain difference in the left and right bow turning velocity of the robot, which is mainly due to manufacturing errors of the robot.

(3) Performance testing of the robot snorkeling

The snorkeling modes of soft bionic robot mainly include two modes: flippers torsion at the same time, same frequency, different offset + s well as tail left-right swing / dorsal-ventral movement movement, and snorkeling can be completed with the assistance of trunk inflation and pumping. Among them, the main control chip collects the pressure value measured by the pressure sensor and calculates the depth of the robot. The motion frequency of the flippers of the robot is 0.8Hz, and the amplitude is 1.05rad, and the left-right swing frequency of the tail is 0.8Hz, and the swing amplitude is 0.61rad. Among them, for upward floating movement, the left flipper torsional offset is 5, the right flipper torsional offset is -5, and the trunk inflation pressure is 130kpa; For diving movement, the left flipper torsional offset is -5, the right flipper torsional offset is 5, and the trunk suction pressure is 130kpa. Collect and process the test data of the robot snorkeling experiment, and the specific results are shown in Table 3.

According to Table 3, the upward floating and diving velocities of the soft bionic robot are 87.2mm/s and 79.4mm/s, respectively. Among them, there is a certain difference in the diving velocity of the robot, mainly due to the different abilities of the robot to inflate and extract air to change the volume density of the trunk.

In summary, the experimental test of swimming performance shows that the pneumatic soft bionic robot has certain maneuverability and environmental adaptability, which further verifies that CPG-ANFIS-PID can achieve the expected control objectives.

## 5 Conclusion

This article researches the motion control strategy of the pneumatic soft bionic robot, achieving the accuracy and anti-interference of multiple motion modes of the robot control, which can provide a reference for the research of soft bionic robot motion control.

1. Analyzed the dynamic characteristics of the Hopfield neuron nonlinear oscillator and established a Hopfield oscillator model. Based on the motion characteristics of the robot

**Table 3. Test results of the soft bionic robot in snorkeling.**

| Experiment form | Depth $s$(mm) | Time $t$(s) | Velocity $v$(mm/s) |
|---|---|---|---|
| Upward floating process | 1500 | 17.2 | 87.2 |
| Diving process | 1500 | 18.9 | 79.4 |

and the driving characteristics of the pneumatic system, the oscillator model has been improved, and the coupling coefficients between the main executing mechanisms and the input parameters of the oscillator amplitude have been adjusted. A robot CPG motion control system model was established using the proposed two-layer CPG network topology. By adjusting control parameters through simulation, effective control and smooth switching of multiple motion modes of the robot are achieved.

2. Introducing ANFIS theory and learning algorithms, an 18-parallel two-input single-output ANFIS-PID control structure was established. By setting fuzzy states and dynamically adjusting correction parameters, the membership function after offline learning and training was obtained. The simulation platform was used to simulate different swimming modes of the robot, indicating that the CPG-ANFIS-PID control system can compensate for CPG motion control system errors and has anti-interference performance.

3. The experimental test of the pneumatic soft bionic robot is completed, and the CPG-ANFIS-PID control algorithm is proved to be superior in the performance test of straight swimming at low velocity and medium velocity. In the swimming performance test of straight swimming, turning bow, and snorkeling underwater, it is determined that CPG-ANFIS-PID can achieve the expected control purpose and make the robot have certain maneuverability, flexibility, and environmental adaptability.

## Supporting information

**S1 File. The minimal data set.**
(DOCX)

## Author Contributions

**Conceptualization:** Wenchuan Zhao, Lijian Yang, Ning Wang.

**Data curation:** Wenchuan Zhao, Yu Zhang, Linghui Peng.

**Formal analysis:** Wenchuan Zhao, Yu Zhang, Kian Meng Lim.

**Funding acquisition:** Yu Zhang.

**Investigation:** Wenchuan Zhao, Yu Zhang, Lijian Yang, Ning Wang.

**Methodology:** Wenchuan Zhao, Yu Zhang, Kian Meng Lim, Lijian Yang.

**Project administration:** Wenchuan Zhao, Yu Zhang.

**Resources:** Wenchuan Zhao, Yu Zhang, Kian Meng Lim.

**Software:** Wenchuan Zhao, Ning Wang, Linghui Peng.

**Supervision:** Wenchuan Zhao, Yu Zhang, Kian Meng Lim, Lijian Yang.

**Validation:** Wenchuan Zhao, Yu Zhang, Kian Meng Lim, Lijian Yang, Ning Wang, Linghui Peng.

**Visualization:** Wenchuan Zhao, Ning Wang.

**Writing – original draft:** Wenchuan Zhao.

**Writing – review & editing:** Wenchuan Zhao, Yu Zhang, Kian Meng Lim, Lijian Yang, Linghui Peng.

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
