## [Decision Letter · Decision Letter 0]

26 Mar 2024

PONE-D-24-04796Research on Control Strategy of Pneumatic Soft Bionic Robot Based on Improved CPGPLOS ONE

Dear Dr. Zhao,

Thank you for submitting your manuscript to PLOS ONE. After careful consideration, we feel that it has merit but does not fully meet PLOS ONE’s publication criteria as it currently stands. Therefore, we invite you to submit a revised version of the manuscript that addresses the points raised during the review process.

We look forward to receiving your revised manuscript.

Kind regards,

Van Thanh Tien Nguyen, Ph.D.

Academic Editor

PLOS ONE

“This work was done in the Institute of Robotics at Shenyang University of Technology with support from the National Natural Science Foundation of China (Grant No. 52005344). We are grateful for this support. Besides, we would also like to thank the Department of Mechanical Engineering of the National University of Singapore for their support for the develop successfully of this work.”

5. We note that your Data Availability Statement is currently as follows: [All relevant data are within the manuscript and its Supporting Information files.]

Additional Editor Comments:

Dear Dr. Wenchuan Zhao,

The review stage has been completed, and we are pleased to inform you that we have received positive feedback from the reviewers regarding your work.

We kindly request that you revise your manuscript in accordance with the questions and comments raised by the reviewers. It is essential that you provide a clear, point-to-point response to each reviewer. Additionally, please meticulously review the English grammar and structure in your revised manuscript.

Once you have completed the revisions, please resubmit the manuscript, and we will proceed with the processing.

Please do not hesitate to reach out if you have any further questions or concerns.

Thank you for your valuable contributions.

Sincerely yours,

Reviewers' comments:

Reviewer's Responses to Questions

**Comments to the Author**

1. Is the manuscript technically sound, and do the data support the conclusions?

Reviewer #1: Yes

Reviewer #2: Partly

2. Has the statistical analysis been performed appropriately and rigorously? 

Reviewer #1: Yes

Reviewer #2: Yes

3. Have the authors made all data underlying the findings in their manuscript fully available?

Reviewer #1: No

Reviewer #2: Yes

4. Is the manuscript presented in an intelligible fashion and written in standard English?

Reviewer #1: Yes

Reviewer #2: Yes

5. Review Comments to the Author

Reviewer #1: In this paper, the motion control strategy of pneumatic soft bionic robot based on improved central pattern generator (CPG) and adaptive neuro-fuzzy inference system (ANFIS) is proposed. By constructing a two-layer neural network model coupled by 22 Hopfield neuron oscillators, the motion of the robot is effectively controlled. The simulation and experimental results show that the strategy can quickly respond to the change of input parameters and output stable position signals. The strategy successfully realizes the propulsion waveform of biological rhythm movement, which shows its accuracy and anti-interference characteristics. Generally speaking, this research establishes a novel control strategy for pneumatic soft bionic robot, and contributes to the motion control in soft robot. However, before the paper is accepted, the author should consider the following questions.

1. In the first part of this paper, many researches on robot control based on cpg are introduced. Now the existing researches have realized adaptive control, which can improve the performance. What are the obvious advantages of the motion control system proposed in this paper compared with it?

2. How did the CPG control model expression (1) in Section 2.1 of the article come from? Please explain.

3. In the establishment of the topological structure in Section 2.4, there is no coupling relationship between the nodes in the first layer and the second layer. When the nodes in the first layer and the nodes in the second layer make mistakes, the signal transmission between them can't be realized, so it seems that the control system can't adjust and correct them by other nodes. Please give a reasonable explanation and a solution.

4. For the experimental test, the paper only compares the signal data of neurons with the simulation data, and whether the collected data such as motion trajectory and swimming distance mentioned in the article can be given, so as to intuitively see whether the control system can achieve the expected motion goal.

5. In the conclusion of this paper, it is mentioned that the motion control strategy realizes the accuracy and anti-interference of various motion modes controlled by the robot. What is the anti-interference here? Please introduce it in detail.

Reviewer #2: This paper proposed a CPG-ANFIS-PID motion control strategy for a pneumatic soft bionic robot, which can quickly respond to input parameter changes between different swimming modes, and stably output smooth and continuous dynamic position signals. However, the verification process for control effectiveness in the paper is relatively brief, and here are some suggestions:

1. The experimental verification section is too simple, only providing information on the output signal. It is necessary to supplement the actual control effect of the proposed control algorithm on the robot, including but not limited to swimming speed, turning radius, and the switching effect of different motions.

2. As authors state, “This article combines the complementary characteristics of neural networks and fuzzy theory to build an ANFIS, which can fully utilize the learning ability of neural networks and the ability of fuzzy systems to deal with fuzzy and qualitative knowledge.” (Page 14). The authors need to add comparisons in simulation and experiments to demonstrate the advantages of CPG-ANFIS-PID motion control strategy, such as the comparison results with general CPG control.

3. The introduction section of the paper is simply a stack of references and requires further logical enhancement.

4. References on CPG control are not comprehensive enough, here are some examples:

[1] Thandiackal, R., Melo, K., Paez, L., Herault, J., Kano, T., Akiyama, K., … & Ijspeert, A. J. (2021). Emergence of robust self-organized undulatory swimming based on local hydrodynamic force sensing. Science robotics, 6(57), eabf6354.

[2] Zhong, Y., Hong, Z., Li, Y., & Yu, J. (2023). A General Kinematic Model of Fish Locomotion Enables Robot Fish to Master Multiple Swimming Motions. IEEE Transactions on Robotics.

[3] Ijspeert A J, Crespi A, Ryczko D, et al. From swimming to walking with a salamander robot driven by a spinal cord model[J]. science, 2007, 315(5817): 1416-1420.

[4] Yu, J., Wu, Z., Wang, M., & Tan, M. (2015). CPG network optimization for a biomimetic robotic fish via PSO. IEEE transactions on neural networks and learning systems, 27(9), 1962-1968.

5. The clarity of the image should be further improved, preferably by using vector graphics.

6. PLOS authors have the option to publish the peer review history of their article (what does this mean?). If published, this will include your full peer review and any attached files.

Reviewer #1: No

Reviewer #2: No

---

## [Author Response · Author response to Decision Letter 0]

16 May 2024

Thank you very much for comments. We have strictly revised the manuscript according to the requirements of the reviewers. However, due to the inclusion of images and formulas in the reply, it cannot be fully copied into this column. We have uploaded "Response to Reviewers" to "Attach Files" for your review. If you have any questions, please feel free to contact us. Thank you very much.

---

## [Decision Letter · Decision Letter 1]

6 Jun 2024

PONE-D-24-04796R1Research on control strategy of pneumatic soft bionic robot based on improved CPGPLOS ONE

Dear Dr. Zhao,

Thank you for submitting your manuscript to PLOS ONE. After careful consideration, we feel that it has merit but does not fully meet PLOS ONE’s publication criteria as it currently stands. Therefore, we invite you to submit a revised version of the manuscript that addresses the points raised during the review process.

We look forward to receiving your revised manuscript.

Kind regards,

Van Thanh Tien Nguyen, Ph.D.

Academic Editor

PLOS ONE

Journal Requirements:

Additional Editor Comments:

Please note that the reviewers' articles suggestion for citing in your manuscript are not encouraged if they are not related to your work. Thạnk you. 

Reviewers' comments:

Reviewer's Responses to Questions

**Comments to the Author**

1. If the authors have adequately addressed your comments raised in a previous round of review and you feel that this manuscript is now acceptable for publication, you may indicate that here to bypass the “Comments to the Author” section, enter your conflict of interest statement in the “Confidential to Editor” section, and submit your "Accept" recommendation.

Reviewer #1: All comments have been addressed

Reviewer #2: All comments have been addressed

2. Is the manuscript technically sound, and do the data support the conclusions?

Reviewer #1: Yes

Reviewer #2: Yes

3. Has the statistical analysis been performed appropriately and rigorously? 

Reviewer #1: Yes

Reviewer #2: Yes

4. Have the authors made all data underlying the findings in their manuscript fully available?

Reviewer #1: Yes

Reviewer #2: Yes

5. Is the manuscript presented in an intelligible fashion and written in standard English?

Reviewer #1: Yes

Reviewer #2: Yes

6. Review Comments to the Author

**Reviewer #1:** It can be clearly seen that the author has put a lot of efforts into integrating the opinions and suggestions put forward in the first review. The current version of this paper is more comprehensive. The motion control strategy of pneumatic soft bionic robot based on improved central pattern generator (CPG) and adaptive neuro-fuzzy inference system (ANFIS) realizes accurate control and various tests, which provides valuable resources for the control of multi-motion modes of pneumatic flexible bionic robot with complex structure.

The references can be further expanded to highlight the significance of this study. Such and other relative references can be referred.

[1] The neuromechanics of animal locomotion: From biology to robotics and back

[2] Target tracking control of a bionic mantis shrimp robot with closed-loop central pattern generators

[3] Development and Motion Control of Biomimetic Underwater Robots: A Survey

After the above revisions, the manuscript can be accepted for publication.

**Reviewer #2:** This paper mainly discusses how to improve the motion control precision and anti-interference ability of soft robot, and puts forward the control strategy suitable for pneumatic soft bionic robot. The English writing is well and all figures are clear. Moreover, the authors have addressed my comments. I have no more question and I think it can be accept in current form. In the future, the authors can consider how to do a better integration without tether.

7. PLOS authors have the option to publish the peer review history of their article (what does this mean?). If published, this will include your full peer review and any attached files.

Reviewer #1: **Yes: **Gang Chen

Reviewer #2: No

---

## [Author Response · Author response to Decision Letter 1]

6 Jun 2024

Journal Requirements:

Response: Thank you very much for your comments. All content is correct, and there is no citation of the withdrawn article.

Additional Editor Comments:

Please note that the reviewers' articles suggestion for citing in your manuscript are not encouraged if they are not related to your work. Thạnk you. 

Response: Thank you very much for your comments. The articles recommended by the reviewer are relevant to our research work and has been cited.

Review Comments to the Author

Reviewer #1: It can be clearly seen that the author has put a lot of efforts into integrating the opinions and suggestions put forward in the first review. The current version of this paper is more comprehensive. The motion control strategy of pneumatic soft bionic robot based on improved central pattern generator (CPG) and adaptive neuro-fuzzy inference system (ANFIS) realizes accurate control and various tests, which provides valuable resources for the control of multi-motion modes of pneumatic flexible bionic robot with complex structure.

The references can be further expanded to highlight the significance of this study. Such and other relative references can be referred.

[1] The neuromechanics of animal locomotion: From biology to robotics and back.

[2] Target tracking control of a bionic mantis shrimp robot with closed-loop central pattern generators.

[3] Development and Motion Control of Biomimetic Underwater Robots: A Survey

Response: Thank you very much for your recognition and support of our achievements. We have carefully studied the 3 articles you recommended, and the content of the articles is very valuable for reference. We cited [7], [13], and [41] in Reference. Among them, the reference [41] is discussed in the second to last paragraph of the Introduction, and the revision is located in the third to seventh lines on page 4.

.

Reviewer #2: This paper mainly discusses how to improve the motion control precision and anti-interference ability of soft robot, and puts forward the control strategy suitable for pneumatic soft bionic robot. The English writing is well and all figures are clear. Moreover, the authors have addressed my comments. I have no more question and I think it can be accept in current form. In the future, the authors can consider how to do a better integration without tether.

Response: Thank you very much for your recognition and support of our achievements.

---

## [Editor Report · Decision Letter 2]

16 Jun 2024

Research on control strategy of pneumatic soft bionic robot based on improved CPG

PONE-D-24-04796R2

Dear Dr. Zhao,

We’re pleased to inform you that your manuscript has been judged scientifically suitable for publication and will be formally accepted for publication once it meets all outstanding technical requirements.

Kind regards,

Van Thanh Tien Nguyen, Ph.D.

Academic Editor

PLOS ONE
---

## [Editor Report · Acceptance letter]

24 Jun 2024

PONE-D-24-04796R2 

PLOS ONE

Dear Dr. Zhao, 

I'm pleased to inform you that your manuscript has been deemed suitable for publication in PLOS ONE. Congratulations! Your manuscript is now being handed over to our production team.

Kind regards, 

on behalf of

Asst. Prof. Van Thanh Tien Nguyen 

Academic Editor

PLOS ONE